# Low-Voltage-Driven SnO_2_-Based H_2_S Microsensor with Optimized Micro-Heater for Portable Gas Sensor Applications

**DOI:** 10.3390/mi13101609

**Published:** 2022-09-27

**Authors:** Dong Geon Jung, Junyeop Lee, Jin Beom Kwon, Bohee Maeng, Hee Kyung An, Daewoong Jung

**Affiliations:** Advanced Mechatronics R&D Group, Korea Institute of Industrial Technology (KITECH), Seoul 31056, Korea

**Keywords:** gas sensor, tin oxide, micro-heater, MEMS, hydrogen sulfide

## Abstract

To realize portable gas sensor applications, it is necessary to develop hydrogen sulfide (H_2_S) microsensors capable of operating at lower voltages with high response, good selectivity and stability, and fast response and recovery times. A gas sensor with a high operating voltage (>5 V) is not suitable for portable applications because it demands additional circuitry, such as a charge pump circuit (supply voltage of common circuits is approximately 1.8–5 V). Among H_2_S microsensor components, that is, the substrate, sensing area, electrode, and micro-heater, the proper design of the micro-heater is particularly important, owing to the role of thermal energy in ensuring the efficient detection of H_2_S. This study proposes and develops tin (IV)-oxide (SnO_2_)-based H_2_S microsensors with different geometrically designed embedded micro-heaters. The proposed micro-heaters affect the operating temperature of the H_2_S sensors, and the micro-heater with a rectangular mesh pattern exhibits superior heating performance at a relatively low operating voltage (3–4 V) compared to those with line (5–7 V) and rectangular patterns (3–5 V). Moreover, utilizing a micro-heater with a rectangular mesh pattern, the fabricated SnO_2_-based H_2_S microsensor was driven at a low operating voltage and offered good detection capability at a low H_2_S concentration (0–10 ppm), with a quick response (<51 s) and recovery time (<101 s).

## 1. Introduction

Hydrogen sulfide (H_2_S), which is a toxic, harmful, corrosive, and colorless gas, is produced by oil deposits, as well as biogas and natural gas fields. Thus, developing an H_2_S sensor with excellent performances, such as good response, selectivity, stability, and a fast response and recovery time, is crucial for the health and safety of industrial workers and the general population. With the advent of the internet of things era, high-performance H_2_S sensors driven at a low voltage and low power have been examined with semiconducting metal oxide (SMO) as the sensing material [1,2,3]. In particular, SMOs such as tin dioxide (SnO_2_), zinc oxide (ZnO), tungsten trioxide (WO_3_), nickel oxide (NiO), and copper oxide (CuO) have been identified as the most promising H_2_S sensing materials. Among these, SnO_2_ is most widely utilized as an H_2_S-sensing material in SMO-based gas sensors because of its excellent gas-detection properties (a good compromise between price, stability, and reliability of material, together with a relatively low operating temperature, and a fast response and recovery time) and numerous fabrication advantages; that is, low-cost, simple fabrication, and good compatibility with the micro-electromechanical (MEMS) process [4,5,6,7]. SnO_2_ has been applied to H_2_S gas sensors in various forms, such as thin films, thick films, pellets, and hot-wire type. SnO_2_ is an n-type semiconductor and an H_2_S-sensor, based on its utilization of resistance–change mechanisms wherein there is an induced variation of the depletion region, owing to the adsorption of ionized oxygen species (O_2_^−^, O^−^, and O^2−^) on the SnO_2_ surface, as shown in Figure 1. The oxygen-related gas-sensing mechanism involves the absorption of oxygen molecules on the SnO_2_ surface to generate chemisorbed oxygen species (O_2_^−^, O^−^ and O^2−^) by capturing electrons from the conductance band, which makes the SnO_2_ surface highly resistive.

When the SnO_2_ surface is exposed to a reductive gas (H_2_S), the reductive gas (H_2_S) upon reacting with the oxygen species (O_2_^−^, O^−^, and O^2−^) reduces the concentration of the oxygen species on this surface, thereby increasing the electron concentration [8,9,10,11,12,13,14,15,16,17,18]. Oxygen species with different forms (O_2_^−^, O^−^ and O^2−^), which are adsorbed on the SnO_2_ surface, are reliant on sensing temperature; therefore, controlling the temperature of H_2_S sensor is vital. In general, the adsorption of O_2_^−^ is dominant in the range of 150–200 °C (1), and the adsorption of O^−^ dominates above 200 °C (2). A further increase in temperature above 400 °C tends to result in the domination of the adsorption of O^2−^ (3) [19]. The process flow is detailed as follows:O_2_ (gas) → O_2_ (physisorption) → O_2_^−^ (chemisorption) → 2O^−^ (chemisorption)
2H_2_S + 3O_2_^−^ → 2SO_2_ + 2H_2_O + 3*e*^−^(1)
H_2_S + 3O^−^ → SO_2_ + H_2_O + 3*e*(2)
H_2_S + 3O^2−^ → SO_2_ + H_2_O + 6*e*^−^(3)

As mentioned above, the SnO_2_ interacts well with H_2_S in a wide range of temperatures; however, it is not suitable for the selective detection of H_2_S. This is because various reducing gases, such as hydrogen, carbon monoxide, ammonia, and others, interact with SnO_2_ in similar ways. Despite the many advantages of SnO_2_, the pristine SnO_2_ gas sensor usually suffers from poor selectivity. Consequently, diverse effective approaches have been conducted to improve the selectivity of SnO_2_-based gas sensors [19,20], such as noble metal doping, composite hetero-structure design, and controlling the reaction temperature. Among these methods, controlling the optimal reaction temperature is the simplest and most effective method. In general, pristine SnO_2_ has an excellent response and a good selectivity at 150–200 °C for H_2_S. Thus, developing an SnO_2_-based H_2_S microsensor which operates well at 150–200 °C is important [21,22,23]. Microsensors for detecting H_2_S comprise a micro-heater, inter-digitated electrode (IDE), and sensing material. The micro-heater (which elevates temperature) embedded in the gas sensor has an important role to play, as aforementioned, in improving the performance of an H_2_S microsensor. It supplies sufficient thermal energy for the reaction between the target gas (H_2_S), oxygen species (O_2_^−^, O^−^ and O^2−^), and sensing material, thereby boosting the H_2_S microsensor performance. To apply the fabricated H_2_S microsensor with a built-in micro-heater to portable applications, a well-designed sensor interface circuit that can supply an appropriate voltage to the sensor is essential. Common sensor interface circuits utilized in commercial portable application supply voltage in the range of 1.8–5 V [24,25,26]. However, additional circuitry, such as a charge pump circuit, is required to supply a high voltage when an H_2_S microsensor with a high operating voltage is used. This results in additional power consumption and a larger footprint. Therefore, it is important to develop a low-voltage-driven H_2_S microsensor. In this study, low-voltage-driven SnO_2_-based H_2_S microsensors with an optimized micro-heater were designed, fabricated, and characterized based on experimental requirements. To investigate the relationship between the H_2_S-detection performance and the heating performance influenced by the geometric design of the micro-heater, micro-heaters with different patterns were fabricated in the proposed H_2_S microsensor platform and characterized. Finally, a low-voltage-driven (3–4 V) SnO_2_-based H_2_S microsensor with an optimized micro-heater was developed, and it was used to detect H_2_S at a low concentration.

## 2. Design and Fabrication of Micro-Heater Embedded in SnO_2_-Based H_2_S Microsensor

The performance of a micro-heater utilizing Joule heating is affected by various factors, such as electrical, mechanical, and material properties, as well as its geometric design. Materials used for a micro-heater are primarily metallic because of their high electrical conductivity, satisfactory specific heat capacity, and good compatibility with the MEMS process. Recently, with the majority of gas sensors being minimized for real-time monitoring and portable applications, the area wherein the micro-heater is fabricated has been limited and minimized as well. Therefore, an optimal geometric design of a micro-heater is certain to improve the heating performance; developing such a design, with an excellent heating performance in a small area, is critical. In this study, SnO_2_-based H_2_S microsensors with three types of micro-heaters were proposed and designed as shown in Figure 2.

The micro-heater types #1–#3 had patterns of meander, rectangular, and rectangular mesh, respectively. The proposed SnO_2_-based H_2_S microsensor comprised micro-heaters (types #1–#3), a temperature sensor, an IDE, and a sensing material (SnO_2_). The sensor and sensing area measured 3 mm × 3 mm and 100 μm × 100 μm, respectively. The width and thickness of a micro-heater, temperature sensor, and IDE were 20 μm and 200 nm, respectively. To minimize the loss of thermal energy produced by micro-heaters, a quartz wafer was used as the sensor substrate. Platinum (Pt), which exhibits a linear relationship between temperature and resistance, was used to fabricate the micro-heater and temperature sensor. Gold (Au) and SnO_2_ with thickness of 50.3 nm each were utilized as the IDE and sensing material, respectively.

Figure 3 shows the fabrication process of the proposed SnO_2_-based H_2_S microsensor. First, the quartz wafer (sensor substrate) was cleaned with an acetone and methanol solution for 10 min. Then, the proposed micro-heaters of three types and the temperature sensor were fabricated through photolithography (for the patterning of the desired geometric designs) and e-beam evaporation (Pt deposition) processes. Silicon nitride (Si_3_N_4_) was deposited via a plasma-enhanced chemical-vapor-deposition process. Subsequently, the deposited Si_3_N_4_ was used for electrical insulation and passivation. The IDE was fabricated through photolithography and e-beam evaporation processes for Au deposition. Finally, SnO_2_, used as the H_2_S-sensing material, was deposited via a sputtering process, and Si_3_N_4_ was etched to fabricate the electrical pads of the micro-heaters and the temperature sensor. Figure 4a,b show the fabricated SnO_2_-based H_2_S microsensor.

## 3. Characterization of Micro-Heater Embedded in SnO_2_-Based H_2_S Microsensor

The fabricated sensing material (SnO_2_) for detecting H_2_S was examined via X-ray diffraction (XRD), and the XRD curves are shown in Figure 5. The results of the diffraction peaks in the 2-theta range from 10° to 90° verified that the crystal structure of the sensing material (SnO_2_) with a varying thickness is the standard tetragonal–rutile crystal phase of SnO_2_. The observed peaks of the deposited SnO_2_ matched well with the standard JCPDS data of SnO_2_.

The performance of the fabricated temperature sensor and micro-heaters with different geometric designs was verified before the SnO_2_-based H_2_S microsensor was characterized, along with the H_2_S concentration. The fabricated temperature sensor’s resistance was measured by modulating the gas chamber temperature. The measured resistance values of the temperature sensor at 30, 100, 200, and 300 °C were approximately 70, 83, 101, and 115 ohm, respectively. These measured resistance values of the temperature sensor were linearly increased by increasing the gas chamber temperature, as shown in Figure 6a. This implied that the heating performance of the micro-heaters embedded in the H_2_S microsensor could be estimated in real time. Next, various input voltage values were applied to the micro-heaters and their heating performance was characterized by measuring the resistance of the temperature sensor (closely fabricated to the micro-heaters). The fabricated micro-heaters of types #1–#3 exhibited different heating performances, implying that the generated thermal energy differed based on the geometric design of a micro-heater for the same input voltage applied to the micro-heaters, as shown in Figure 6b. As the heating performance of a micro-heater was affected by Joule heating, which is closely related to the current traveling through the micro-heater, increasing the current traveling through the micro-heater was very important. The measured initial resistance value of micro-heaters of types #1–#3 were 107.44, 30.27, and 22.35 ohm, respectively. The micro-heater of type #3 exhibited the lowest initial resistance value, thereby enabling a greater flow of current at an equal input voltage of the micro-heater. Therefore, the micro-heater with the rectangular mesh pattern (Type #3) produced more heat energy than the others employing different patterns (Type #1–#2) (for the equal input voltage value being applied). Thus, the micro-heater with the rectangular mesh pattern can produce thermal energy effectively. Simultaneously, the heating performance, along with the micro-heater’s pattern, was also confirmed by estimating the H_2_S-detection performance of the SnO_2_-based H_2_S microsensor.

To characterize the fabricated SnO_2_-based H_2_S microsensor with micro-heaters (types #1–#3), it was placed in the prepared gas chamber and H_2_S gas was injected at a concentration of 0 to 10 ppm. The operating temperature of the H_2_S microsensor can be estimated via the measured resistance of the temperature sensor, along with the input voltage applied to micro-heaters. Table 1 presents the expected operating temperature and power consumption along with the input voltage applied to micro-heaters.

The expected operating temperature was derived by using the relationship between the temperature and measured resistance of temperature sensor, whereas expected power consumption was derived using the power consumption formula (P = V^2^/R). The temperature at which an excellent performance (high response and good selectivity) of SnO_2_ for H_2_S is ensured is 150–200 °C, as mentioned above. Therefore, the input voltage was applied to the fabricated micro-heaters (#1–#3) to elevate the optimal temperature (150–200 °C) and initiate a reaction between H_2_S and SnO_2_. The resistance of SnO_2_ used as an H_2_S-sensing material changed when it was exposed to H_2_S, as mentioned above. The output current of the fabricated H_2_S microsensor was measured by injecting H_2_S gas in the range of 0 to 10 ppm, as shown in Figure 7.

The variation in the output current was closely related to the chemical properties of the fabricated SnO_2_ surface oxygen. Oxygen was absorbed on the SnO_2_ surface in different forms depending on the operating temperature, and it was converted into molecular (physisorption) or dissociative (chemisorption) forms by the increasing operating temperature. The oxygen species with different forms (O_2_^−^, O^−^ and O^2−^) generated on the SnO_2_ surface induced an electron-depletion layer, resulting in the decrease in carrier concentration and increase in resistance on the SnO_2_ surface. Output currents of SnO_2_-based H_2_S microsensors increased when H_2_S was injected into the chamber because the oxygen species adsorbed on the SnO_2_-sensing material surface were consumed by the chemical reaction and the electrons donated back to the SnO_2_ surface, resulting in a decreased electrical resistance. The response of an SnO_2_-based H_2_S microsensor is typically defined as
*S(Response)* = *R*_*air*_*/R*_*gas*_ = *I*_*gas*_*/I*_*air*_,(4)
where *R_air_* and *R_gas_* are the resistances and *I_air_* and *I_gas_* are the conductance values of the sensor regarding air and reducing gas (H_2_S), respectively. The response dramatically improved when the operating temperature was increased by increasing the input voltage of the micro-heater. This is because increasing the operating temperature causes the oxidation of several H_2_S molecules by producing a multitude of electrons. Therefore, the output current varies greatly, thus indicating a significant improvement in the response for H_2_S detection. However, the response is not constantly improved by the increasing operating temperature. In Figure 7b–d, the measured response of the fabricated H_2_S microsensor slightly increased or saturated because the oxygen species were desorbed from the SnO_2_ surface [27]. In addition, at a higher operating temperature, the carrier concentration increased, owing to the intrinsic thermal excitation while the Debye length decreases. These are primarily responsible for the decrease in gas response at higher temperatures [28]. The micro-heater of type #3 exhibited a superior H_2_S-detecting performance at a lower operating voltage compared to micro-heaters of type #1 and #2 because of its lower resistance value, which aided in increasing the current traveling through micro-heater type #3. Thus, the micro-heater of type #3 operated at a relatively lower input voltage, had a superior heating performance, and actively provided sufficient energy for the reaction between H_2_S and the oxygen species (O_2_^−^, O^−^, and O^2−^) on the SnO_2_ surface.

Figure 8a–c show the measured response and recovery time of the H_2_S microsensor for micro-heaters of types #1–#3. The response time is defined as the time required for decreasing the H_2_S sensor resistance or increasing the H_2_S sensor conductance by 90% of the total decrease *(R_air_-R_gas_*) or total increase (*I_gas_-I_air_*). In contrast, the recovery time is defined as the time required to recover the H_2_S sensor resistance or increase the H_2_S sensor conductance by 90% of the total decrease *(R_air_-R_gas_*) or total increase (*I_gas_-I_air_*) when the H_2_S injection is stopped and air is injected into the chamber.

Response and recovery times considerably decreased with the increase in operating temperature, which is controlled by the input voltage of the micro-heater. At a lower input voltage, the micro-heater of type #3 had a shorter response and recovery time compared to that of type #1 and #2, as it produced a more adequate thermal energy for an active and rapid reaction between H_2_S and oxygen species (O_2_^−^, O^−^ and O^2−^) on the SnO_2_ surface, owing to its superior heating ability. To initiate a reaction between molecules, they must be close to each other, and each molecule must have an energy greater than the energy required for the reaction (i.e., activation energy, *Ea*). The activation energy for the reaction between H_2_S and the surface-adsorbed oxygen species (O_2_^−^, O^−^ and O^2−^) decreased with the increasing operating temperature. This decreased activation energy rapidly induced the reaction between H_2_S and oxygen species (O_2_^−^, O^−^ and O^2−^) with an increasing operating temperature. It was confirmed that the fabricated SnO_2_-based H_2_S microsensors with micro-heaters of type #1–#3 actively react with H_2_S at the operating temperature of approximately 170–180 °C; however, the SnO_2_-based H_2_S microsensor’s response for H_2_S was saturated or degraded when the operating temperature was further increased above 180 °C, as shown in Figure 9. In the high response region, the SnO_2_-based H_2_S microsensor with a micro-heater of type #3 exhibited a low driven voltage (3.5 V) and a low power consumption (340.65 mW), while offering a high performance (response: 6.52, response time: 51 s, recovery time: 101 s) for portable gas sensor applications, as shown in Figure 9. However, other H_2_S microsensors with micro-heaters of type #1–#2 must be supplied with a higher operating voltage and require a greater power consumption to exhibit a similar H_2_S-detection ability. This limits their applicability in portable gas sensor applications.

To develop an H_2_S gas sensor that requires a lower-driven voltage and power consumption than the developed H_2_S microsensor, a membrane structure must be fabricated. This must be considered depending on applicable fields, as a gas sensor with a membrane structure can incur high costs, encounter difficulties in fabrication, and possess weak mechanical properties.

For practical applications, gas sensors should exhibit a strong response, as well as good selectivity, toward the targeted gas (H_2_S in this study). To estimate the selectivity of the proposed SnO_2_-based H_2_S microsensor with micro-heater type #3 (input voltage is 3.5 V, power consumption is 340.65 mW), the fabricated H_2_S microsensor was exposed to different types of gases, including ammonia (NH_3_), hydrogen (H_2_), and carbon monoxide (CO) gases. As shown in Figure 10, the fabricated H_2_S microsensor displayed great selectivity toward H_2_S. In case of an increasing operating temperature at the higher input voltage (>3.5–4 V), the fabricated H_2_S microsensor with the micro-heater of type #3 actively reacted with different types of gases, as mentioned in Section 1. Thus, the selectivity of fabricated H_2_S microsensors is poor, and unnecessary power consumption is required. Based on the experimental results, we confirmed that it is important to provide sufficient thermal energy to reach an optimum operating temperature (170–180 °C) for the reaction between H_2_S and oxygen species (O_2_^−^, O^−^ and O^2^^−^) on the SnO_2_ surface using the micro-heater that exhibits a superior heating performance. 

In summary, the H_2_S-detection ability of the proposed SnO_2_-based H_2_S microsensor can be significantly improved by supplying thermal energy, utilizing fabricated micro-heaters embedded in the H_2_S sensor. Further, the applied input voltage and power consumption can be minimized by optimizing the micro-heater design.

## 4. Conclusions

This study proposed and fabricated an SnO_2_-based H_2_S microsensor with micro-heaters of different geometric designs. An H_2_S sensor using semiconducting metal oxide as the sensing material generally comprises a substrate, sensing material, IDE, and a micro-heater. The micro-heater, embedded in the gas sensor, has an important role to play because the reaction between H_2_S and oxygen species (O_2_^−^, O^−^ and O^2^^−^) on the SnO_2_ surface is affected by the operation temperature of the sensor. The development of a micro-heater producing more thermal energy by minimizing the operating voltage and power consumption is necessary to realize more viable real-time monitoring and portable sensor applications. To meet this requirement, micro-heaters with different geometric designs (meander, rectangular, and rectangular mesh patterns) have been proposed, and their heating performances were characterized by estimating the H_2_S-detection ability of the sensor. This was accomplished by applying a sufficient input voltage and then measuring the resistance of the temperature sensor. Based on the experimental results, we confirmed that the micro-heater with a rectangular mesh pattern produced thermal energy more effectively. Therefore, the SnO_2_-based H_2_S microsensor with the micro-heater with a rectangular mesh pattern displayed a superior H_2_S-detection ability. Its responses (*I_gas_/I_air_*) were 4.37 (2 ppm), 5.64 (4 ppm), 6.56 (6 ppm), 7.31 (8 ppm), and 8.72 (10 ppm) at an applied input voltage of 3.5 V to the micro-heater. Furthermore, it had a shorter response time (<51 s) and recovery time (<101 s) compared to H_2_S microsensors with micro-heaters with meander and rectangular patterns. H_2_S is a toxic and harmful gas, even at concentrations as low as hundreds of parts per million, and is mainly produced by oil deposits, biogas, and natural gas fields. Thus, developing an H_2_S sensor with good selectivity and a fast response time is crucial for the health and safety of industrial workers and the general population. Therefore, the developed and optimized H_2_S sensor proposed in this study is suitable for practical real-time monitoring and portable sensor applications.

## Figures and Tables

**Figure 1 micromachines-13-01609-f001:**
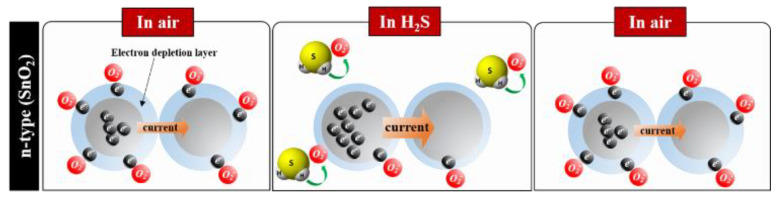
H_2_S-sensing mechanism of SnO_2_.

**Figure 2 micromachines-13-01609-f002:**
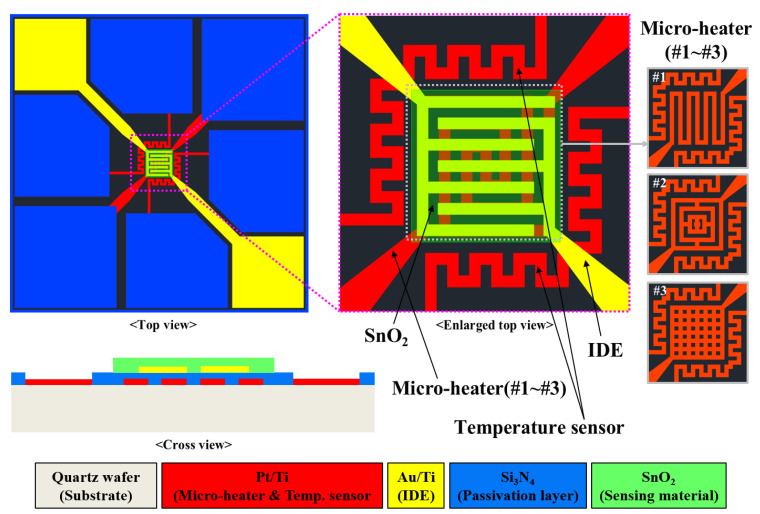
Schematic of the proposed SnO_2_-based H_2_S microsensor with three types of micro-heaters.

**Figure 3 micromachines-13-01609-f003:**
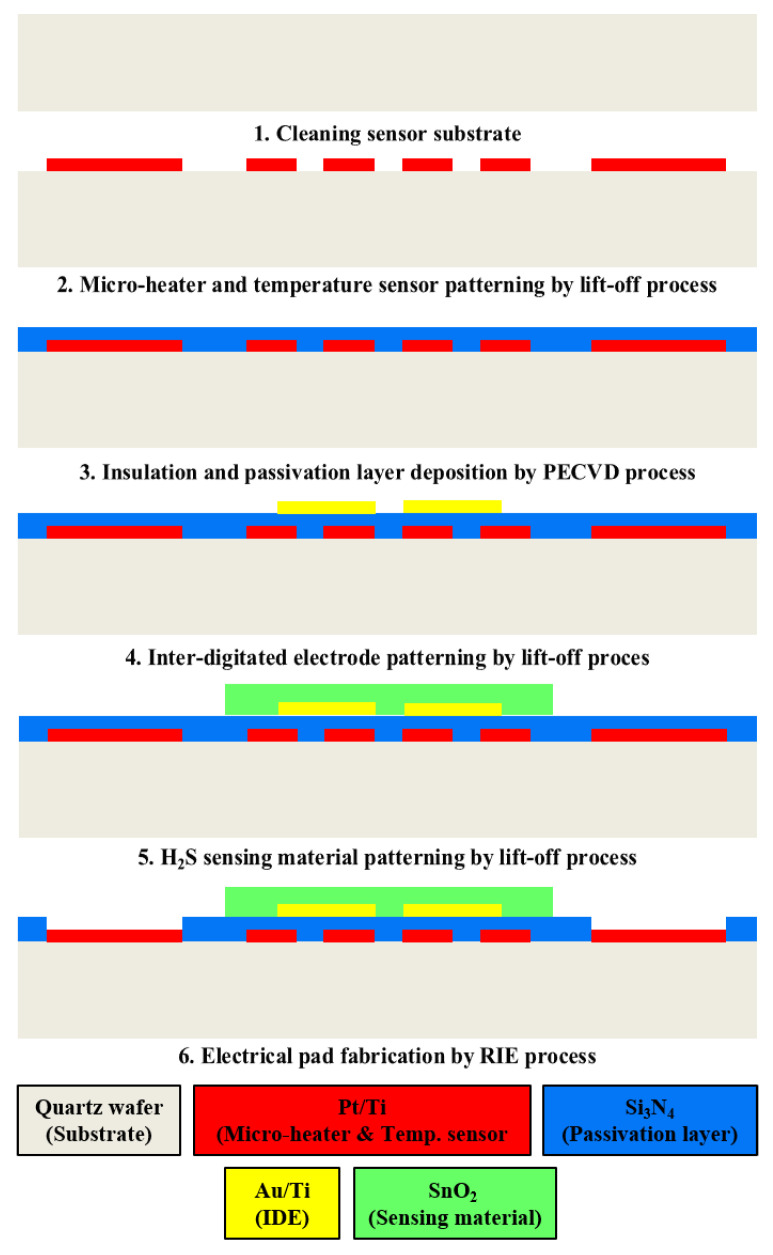
Fabrication process of the proposed SnO_2_-based H_2_S microsensor with micro-heaters of three types.

**Figure 4 micromachines-13-01609-f004:**
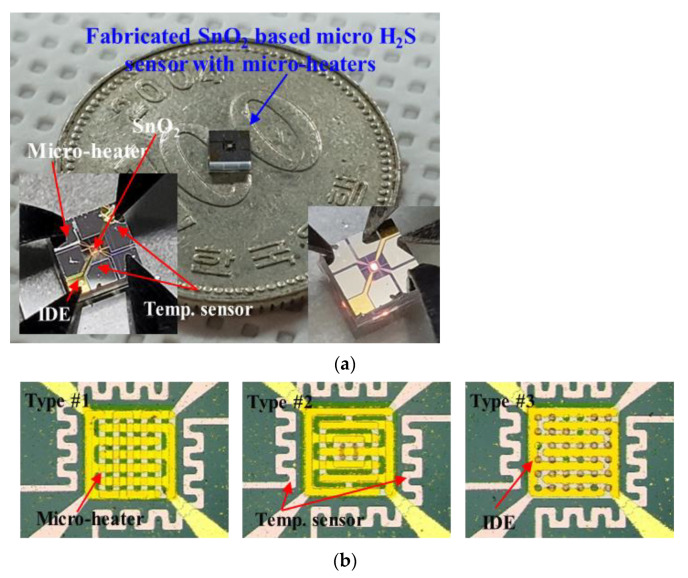
(**a**) Photographic and (**b**) microscopic images of the fabricated SnO_2_-based H_2_S microsensor with three types of micro-heaters.

**Figure 5 micromachines-13-01609-f005:**
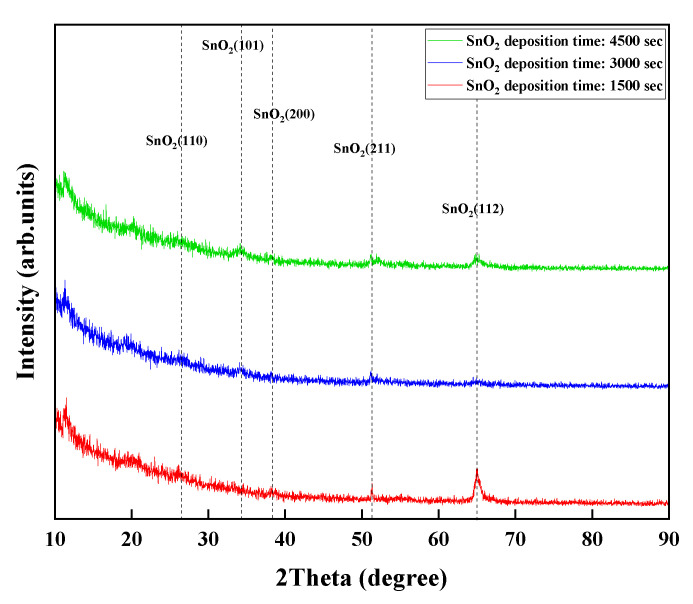
XRD patterns as a function of SnO_2_ depositing time.

**Figure 6 micromachines-13-01609-f006:**
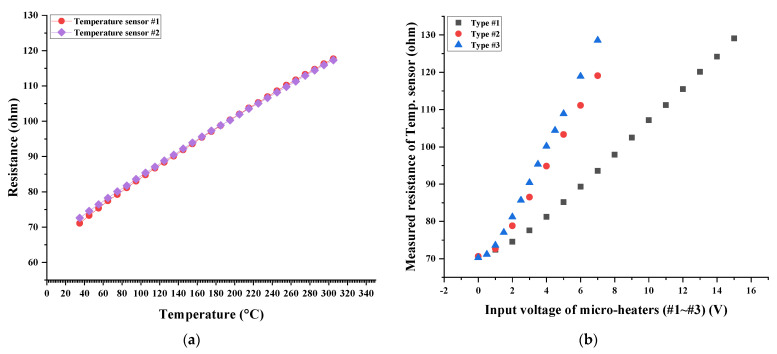
Graphs of (**a**) measured resistance of temperature sensor as a function of temperature change and (**b**) measured resistance of temperature as a function of micro-heater input voltage.

**Figure 7 micromachines-13-01609-f007:**
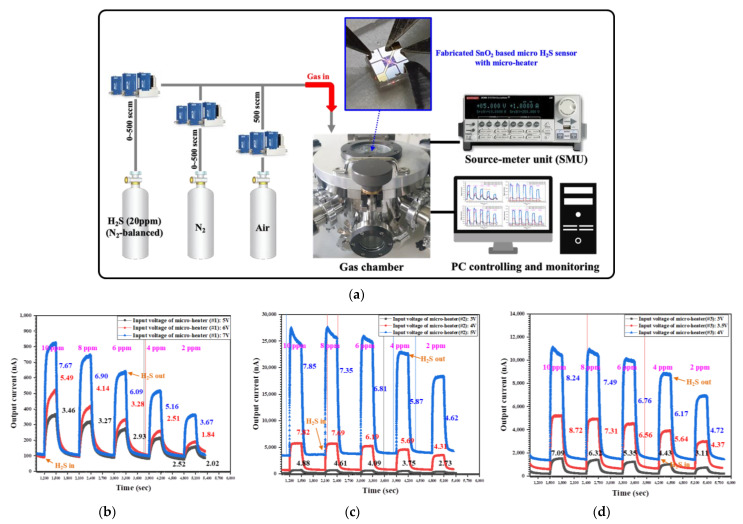
(**a**) Experimental setup for characterization of SnO_2_-based H_2_S microsensor with micro-heaters of different geometric designs, and measured output currents of SnO_2_-based H_2_S microsensor with micro-heaters of (**b**) type #1, (**c**) type #2, and (**d**) type #3 as a function of H_2_S concentration.

**Figure 8 micromachines-13-01609-f008:**
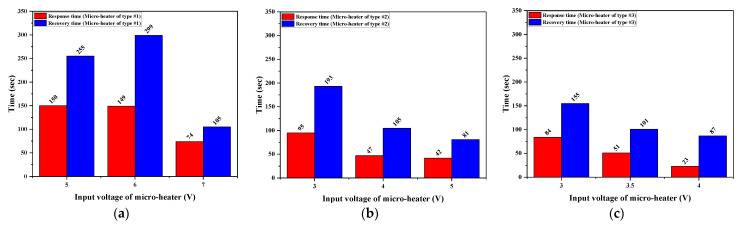
Measured response and recovery time of the SnO_2_-based H_2_S microsensor with micro-heaters of (**a**) type #1, (**b**) type #2, and (**c**) type #3 as a function of input voltage of micro-heater.

**Figure 9 micromachines-13-01609-f009:**
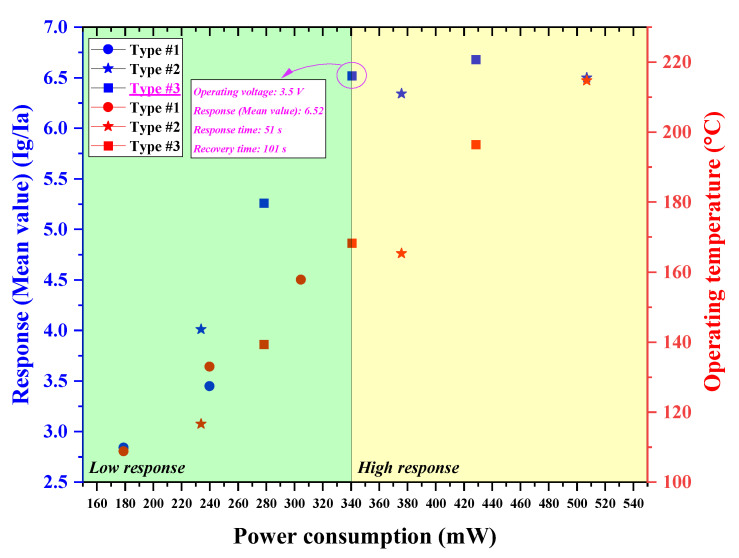
Measured response and expected operating temperature of the SnO2-based H2S microsensor with micro-heaters of type #1–#3 as a function of power consumption.

**Figure 10 micromachines-13-01609-f010:**
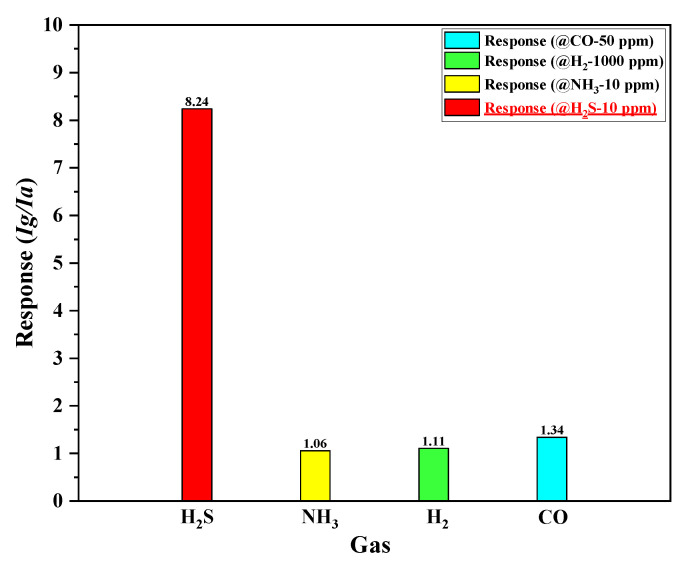
Measured responses of the SnO_2_-based H_2_S microsensor with micro-heater of type #3 to different tested gases.

**Table 1 micromachines-13-01609-t001:** Expected temperature and power consumption of micro-heaters with various geometric designs as a function of the applied input voltage of micro-heaters.

Type of Micro-Heater	Input Voltage Applied to Micro-Heater (V)	Expected Temperature of Micro-Heater (°C)	Expected Power Consumption of Micro-Heater (mW)
Type #1 (meander pattern)	5 V, 6 V, 7 V	108.84 °C, 132.97 °C, 157.8 °C	179 mW, 239.88 mW, 304.5 mW
Type #2 (rectangular pattern)	3 V, 4 V, 5 V	116.59 °C, 165.32 °C, 214.68 °C	233.76 mW, 375.8 mW, 507.05 mW
Type #3 (rectangular mesh pattern)	3 V, 3.5 V, 4 V	139.41 °C, 168.29 °C, 196.47 °C	278.23 mW, 340.65 mW, 428.4 mW

## Data Availability

Not applicable.

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
