# Peer review of "Low-Voltage-Driven SnO2-Based H2S Microsensor with Optimized Micro-Heater for Portable Gas Sensor Applications"

_micromachines, 2022, doi:10.3390/mi13101609_

Round 1

Reviewer 1 Report

This is a succinct report on an experimental study on the Optimization of Microheater Embedded in SnO2 Based H2S Microsensor for Practical Applications which is well presented overall, with clear rationale, methodology and results.  The work is of interest and relevance to the sensors field in which it sits, and the experimental work is well structured.

The final version would benefit from careful proofreading for grammatical construction, but it is generally very well-articulated.

Author Response

This is a succinct report on an experimental study on the Optimization of Microheater Embedded in SnO2 Based H2S Microsensor for Practical Applications which is well presented overall, with clear rationale, methodology and results.  The work is of interest and relevance to the sensors field in which it sits, and the experimental work is well structured.

Q1. The final version would benefit from careful proofreading for grammatical construction, but it is generally very well-articulated.

A1. I entirely revised my paper by reflecting your opinion.

Reviewer 2 Report

The manuscript proposes a microheater geometry (a geometry which isn’t quite new) for chemiresistive H2S sensing with SnO2. They compare this design against known designs.

Major:

- The paper fails entirely to put the work in a context which relates to the current state-of-the-art and the active research in this area. The references, except for one or two, are very old. Furthermore, all the citations are only used to support highly generic statements. These are also lumped in three locations in the paper’s first two paragraphs of the introduction. To me, this suggests tunnel-vision by the authors. Please find out what other groups are doing on this topic and see how your designs compare or fit in with current research.

- In my opinion, the discussion on microheater efficiency is somewhat flawed. The authors seem to suggest that the applied potential is the same as the required power. The authors state “The type #4 microheater had the lowest initial resistance value due to which more current flowed when equal input voltage was applied to the microheaters.” Okay, this is Ohm’s law. Then, they state, “Therefore, the microheater having the rectangular mesh pattern (Type #4) was the most effective in producing thermal energy...” I’m not sure what the authors mean by it being most “effective”, but the microheater power follows P = V^2/R, so an inverse relationship with resistance and square dependence on voltage. A quick calculation of the presented values suggests that the 4 microheater types (from 1 to 4) require the following power to reach 160oC (45.6mW, 52.9mW, 47.8mW, 40.2mW – but only reaches 150oC). If we do a quick extrapolation with the values presented for type 4 at 180oC we get 45.1mW, so slightly better than type 1, but the authors need to include this discussion relating to power and not resistance when they talk about efficiency.

- For the proposed designs, any comparisons to the circular heater are simply unfair. The temperature sensor is placed further away from this circular shape than for the other sensor types. Therefore, I would suggest they exclude this design from their comparison altogether.

- When the authors compare the sensitivity performance, they show different operating condition for different heaters, so it is completely unfair to make any comparative conclusions. The authors should compare the performance for the SAME temperature for each microhetaer. Then, they can make conclusions about sensitivity or efficiency.

- There is no discussion on Figure 7, only results are shown.

- The discussion on page 8 just lists all the results which are presented in the graphs. This makes the paper difficult to read and provides no additional discussion on top of the figures. If the authors really wish to provide the exact numbers for the reader, they should put these in the graphs. This entire paragraph should be removed and the authors should instead discuss the results mean.

- Again, same issue for the “discussion” on Figure 9, the authors just list the results from the figure in the text.

- For Figure 10, also no discussion is provided and no suggestions are made as to why this sensor seems to exhibit a higher sensitivity towards H2S. It seems to be a coincidence or a simple property of the sensing material at this particular temperature.

- Ultimately, the authors conclude with “In summary, H2S detection ability of the proposed SnO2 based H2S microsensor is significantly improved by supplying thermal energy utilizing fabricated microheaters embedded in the H2S sensor; and optimizing a microheater’s heating performance via its geometric design.” However, from everything that they have shown, I do not agree that this is the proper conclusion.

Minor:

- The statement in the abstract, “it is necessary to improve H2S sensing parameters such as response, selectivity, stability, and recovery times by minimizing the power consumption” suggests that minimizing the power consumption will result in improvements in the response, selectivity, etc., which is, of course, not the case. This should be rephrased.

- The authors also make many statements without citing the sources. One example is stating that O2- adsorption is dominant for lower temperatures, while O- adsorption dominates above 200C. This is relatively well known in the community, but nevertheless should be cited in the manuscript.

- The authors refer to microheater #1 as “line”, but this is commonly referred to as “meander” in the community. In fact, it is probably one of the most popular microheater geometries, so it is strange that the authors never use this terminology.

- The authors state “To minimize the loss of thermal energy produced 86 by microheaters, a quartz wafer is used as the sensor substrate.” Usually, the heating element is suspended on a MEMS membrane. Why is quartz preferred by the authors? Using quartz prevents the potential of sensor integration with electronics, which is why silicon and MEMS membranes are commonly desired.

- Fig. 2 shows the IDE layer above the sensor, while Fig. 3 suggests it is the other way around. The authors should clarify if they place the IDE below or above the sensing film. This will also have a large impact on the sensing reaction, since the Schottky contact between the gold and SnO2 will be modified by adsorbed gas molecules.

- The authors should mention all the geometric parameters, such as layer thickness, as well as design rules, specifically relating to the allowed pitch, so the thickness of the metal layers and the space between these.

 - The authors note that “The response dramatically improved when operating temperature was increased by increasing input voltage of microheater.” As it is well known, there is a peak temperature at which the redox reaction is most efficient. This simply means that the authors did not reach that yet.

- The authors state: “According to Maxwell–Boltzmann distribution, as the temperature increases, the number of molecules with energy above the activation energy increases.” This makes the paper seem quite amateurish and more like a lab report than a scientific publication.

Author Response

Reviewer 2

The manuscript proposes a microheater geometry (a geometry which isn’t quite new) for chemiresistive H2S sensing with SnO2. They compare this design against known designs.

Q1. The paper fails entirely to put the work in a context which relates to the current state-of-the-art and the active research in this area. The references, except for one or two, are very old. Furthermore, all the citations are only used to support highly generic statements. These are also lumped in three locations in the paper’s first two paragraphs of the introduction. To me, this suggests tunnel-vision by the authors. Please find out what other groups are doing on this topic and see how your designs compare or fit in with current research.

A1. I added references revised my paper entirely by reflecting your opinion

Q2. In my opinion, the discussion on micro-heater efficiency is somewhat flawed. The authors seem to suggest that the applied potential is the same as the required power. The authors state “The type #4 microheater had the lowest initial resistance value due to which more current flowed when equal input voltage was applied to the microheaters.” Okay, this is Ohm’s law. Then, they state, “Therefore, the microheater having the rectangular mesh pattern (Type #4) was the most effective in producing thermal energy...” I’m not sure what the authors mean by it being most “effective”, but the microheater power follows P = V^2/R, so an inverse relationship with resistance and square dependence on voltage. A quick calculation of the presented values suggests that the 4 microheater types (from 1 to 4) require the following power to reach 160oC (45.6mW, 52.9mW, 47.8mW, 40.2mW – but only reaches 150oC). If we do a quick extrapolation with the values presented for type 4 at 180oC we get 45.1mW, so slightly better than type 1, but the authors need to include this discussion relating to power and not resistance when they talk about efficiency.

A2. I’m sorry that I’ve written my paper vaguely so that others may misunderstood. Purpose of my paper is to optimize micro-heater’s geometric for the portable H2S gas sensor. To commercialize the fabricated gas sensor, sensor interface circuit which supplies operating voltage and amplifies output signal, is essential. Common circuits supply voltage of 1.8~5 V to sensor, and additional circuitry such as a charge pump circuit must be demanded when gas sensor with a high operating voltage is utilized. It causes additional power consumption and a larger footprint. Therefore, developing low-voltage-driven gas sensor is very important and I investigated micro-heater geometric which can travel the larger current flow at low input voltage because joule heating is proportional to I2 and R.

In general, SnO2 has many advantages such as low-cost, good responsivity and compatibility with MEMS process but SnO2 reacts with various gases, actively (selectivity is poor). Therefore, the optimum operating temperature which differs along with target gas must be controlled and H2S has excellent response and good selectivity at 150~200 ℃ (related reference is added). In this respect, I optimizes micro-heater’s geometric which well elevates temperature of 150~200 ℃ by utilizing a low operating voltage. In summary, to effectively elevate the temperature of 150~200 by utilizing a low operating voltage is more important than measuring H2S sensor performance at same temperature. For this purpose, I estimated performance of fabricated H2S sensor with micro-heaters having different geometric by modulating input voltage applied to micro-heater and range of input voltage is set for considering common sensor interface circuit. As a results, SnO2 based H2S micro sensor with micro-heater having rectangular mesh pattern is exhibited the superior H2S detection ability at low-driven-voltage (3.5V) and power consumption (340.65 mW) compare to H2S micro sensors with micro-heater of type #1 and #2. By reflecting your opinions and my purpose to write paper, paper is entirely revised and highlighted in red.         

Q3. For the proposed designs, any comparisons to the circular heater are simply unfair. The temperature sensor is placed further away from this circular shape than for the other sensor types. Therefore, I would suggest they exclude this design from their comparison altogether.

A3. By reflecting your opinion, I exclude the contents related to micro-heater having circular pattern.

Q4. When the authors compare the sensitivity performance, they show different operating condition for different heaters, so it is completely unfair to make any comparative conclusions. The authors should compare the performance for the SAME temperature for each microhetaer. Then, they can make conclusions about sensitivity or efficiency.

A4. As mentioned in A2, I concentrate on the development of micro-heater which elevates temperature of 150~200 ℃ (this temperature is known as the optimum temperature for H2S detection) at low operating voltage for portable sensor applications. Therefore, performance of H2S micro sensor is estimated by modulating input voltage of micro-heater.

Q5. There is no discussion on Figure 7, only results are shown.

A5. Discussion on figure 7 is added and highlighted in red on page 9.

Q6. The discussion on page 8 just lists all the results which are presented in the graphs. This makes the paper difficult to read and provides no additional discussion on top of the figures. If the authors really wish to provide the exact numbers for the reader, they should put these in the graphs. This entire paragraph should be removed and the authors should instead discuss the results mean.

A6. I revised my paper by reflecting your opinion on page 8.

Q7. Again, same issue for the “discussion” on Figure 9, the authors just list the results from the figure in the text.

A7. Discussion on figure 9 is added and highlighted in red on page 9.

Q8 For Figure 10, also no discussion is provided and no suggestions are made as to why this sensor seems to exhibit a higher sensitivity towards H2S. It seems to be a coincidence or a simple property of the sensing material at this particular temperature.

A8. Discussion on figure 9 is added and highlighted in red on page 10.

Q9 Ultimately, the authors conclude with “In summary, H2S detection ability of the proposed SnO2 based H2S microsensor is significantly improved by supplying thermal energy utilizing fabricated microheaters embedded in the H2S sensor; and optimizing a microheater’s heating performance via its geometric design.” However, from everything that they have shown, I do not agree that this is the proper conclusion.

A9. I added and revised paper by reflecting your opinion, these parts were highlighted in red and my opinion also explained in A2.

Minor:

Q1. The statement in the abstract, “it is necessary to improve H2S sensing parameters such as response, selectivity, stability, and recovery times by minimizing the power consumption” suggests that minimizing the power consumption will result in improvements in the response, selectivity, etc., which is, of course, not the case. This should be rephrased.

A1. By reflecting your opinion, i revised and highlighted in red.

Q2. The authors also make many statements without citing the sources. One example is stating that O2- adsorption is dominant for lower temperatures, while O- adsorption dominates above 200C. This is relatively well known in the community, but nevertheless should be cited in the manuscript.

A2. By reflecting your opinion, i added additional references highlighted in red.

Q3. The authors refer to microheater #1 as “line”, but this is commonly referred to as “meander” in the community. In fact, it is probably one of the most popular microheater geometries, so it is strange that the authors never use this terminology.

A3. By reflecting your opinion, i used “meander” instead of “line”

Q4. The authors state “To minimize the loss of thermal energy produced 86 by microheaters, a quartz wafer is used as the sensor substrate.” Usually, the heating element is suspended on a MEMS membrane. Why is quartz preferred by the authors? Using quartz prevents the potential of sensor integration with electronics, which is why silicon and MEMS membranes are commonly desired.

A4. I agree your opinion. But I used quartz substrate which well prevents thermal loss compare to bulk silicon substrate because fabricating MEMS membrane using silicon substrate could be high-cost to fabricate and complex and difficult to develop.  

Q5. Fig. 2 shows the IDE layer above the sensor, while Fig. 3 suggests it is the other way around. The authors should clarify if they place the IDE below or above the sensing film. This will also have a large impact on the sensing reaction, since the Schottky contact between the gold and SnO2 will be modified by adsorbed gas molecules.

A5. IDE is located at below sensing film in cross-view in figure 2.  

Q6. The authors should mention all the geometric parameters, such as layer thickness, as well as design rules, specifically relating to the allowed pitch, so the thickness of the metal layers and the space between these.

A6. By reflecting your opinion, i revised and highlighted in red on page 4.

Q7. The authors note that “The response dramatically improved when operating temperature was increased by increasing input voltage of microheater.” As it is well known, there is a peak temperature at which the redox reaction is most efficient. This simply means that the authors did not reach that yet.

A7. I added the contents related and revised paper by reflecting your opinion, in page 2, 7 and 8.

Q8. The authors state: “According to Maxwell–Boltzmann distribution, as the temperature increases, the number of molecules with energy above the activation energy increases.” This makes the paper seem quite amateurish and more like a lab report than a scientific publication.

A8. I deleted the contents related by reflecting your opinion.

Reviewer 3 Report

Title: Optimization of Microheater Embedded in SnO2 Based H2S Microsensor for Practical Applications.

This work more interesting based on microsensor using SnO2. Also, H2S sensors is more important. authors are developed H2S sensor more relavent gas sensing method. Authors should polish the language of the manuscript.

1. The English of the manuscript needs deeply revision. There are some grammatical and typo errors in the text that need to be re-checked and corrected more carefully.

2. Introduction section: author should be add some more points about "major advantages of SnO2 in this microsensor”. Authors should be check following articles and cite it (https://www.sciencedirect.com/science/article/abs/pii/S0003267020309351, https://www.sciencedirect.com/science/article/pii/S0272884221021258)

3. Figures and captions should be check and verify overall manuscript content.

4. Figure quality not good. Therefore, authors should be improving all.

5. Author must be revising the abstract in this work

Author Response

This work more interesting based on microsensor using SnO2. Also, H2S sensors is more important. authors are developed H2S sensor more relavent gas sensing method. Authors should polish the language of the manuscript.

Q1. The English of the manuscript needs deeply revision. There are some grammatical and typo errors in the text that need to be re-checked and corrected more carefully.

A1. I entirely revised my paper by reflecting your opinion.

Q2. Introduction section: author should be add some more points about "major advantages of SnO2 in this microsensor”. Authors should be check following articles and cite it (https://www.sciencedirect.com/science/article/abs/pii/S0003267020309351, https://www.sciencedirect.com/science/article/pii/S0272884221021258)

A2. I added additional reference by reflecting your opinion.

Q3. Figures and captions should be check and verify overall manuscript content.

A3. I entirely revised my paper by reflecting your opinion.

Q4. Figure quality not good. Therefore, authors should be improving all.

A4. I entirely revised my paper by reflecting your opinion.

Q5. Author must be revising the abstract in this work

A5. I revised abstract in paper by reflecting your opinion.

Round 2

Author Response

Reviewer 2

Q2. In my opinion, the discussion on micro-heater efficiency is somewhat flawed. The authors seem to suggest that the applied potential is the same as the required power. The authors state “The type #4 microheater had the lowest initial resistance value due to which more current flowed when equal input voltage was applied to the microheaters.” Okay, this is Ohm’s law. Then, they state, “Therefore, the microheater having the rectangular mesh pattern (Type #4) was the most effective in producing thermal energy...” I’m not sure what the authors mean by it being most “effective”, but the microheater power follows P = V^2/R, so an inverse relationship with resistance and square dependence on voltage. A quick calculation of the presented values suggests that the 4 microheater types (from 1 to 4) require the following power to reach 160℃ (45.6mW, 52.9mW, 47.8mW, 40.2mW – but only reaches 150℃). If we do a quick extrapolation with the values presented for type 4 at 180℃ we get 45.1mW, so slightly better than type 1, but the authors need to include this discussion relating to power and not resistance when they talk about efficiency

Q2-1(Question in review round 2). While there are now also values of the power consumption, the author does not mention or discuss these in the text. While I understand that the main goal here is to lower the applied bias, the power consumption is nevertheless very important and should be mentioned and discussed.

A2. I added the discussion of power consumption by replacing figure 9 and it is highlighted in blue. Some grammatical and typoerrors in the paper also were corrected.

Q5. There is no discussion on Figure 7, only results are shown.

Q5-1(Question in review round 2). The author simply repeats the values from the graph in the text and has very little discussion on the results. Also, the poor language makes it very difficult to understand.

A5. I added the discussion on figure 7 and it is highlighted in blue on page 8~9. Some grammatical and typoerrors in the paper also were corrected.

Q6. The discussion on page 8 just lists all the results which are presented in the graphs. This makes the paper difficult to read and provides no additional discussion on top of the figures. If the authors really wish to provide the exact numbers for the reader, they should put these in the graphs. This entire paragraph should be removed and the authors should instead discuss the results mean.

Q6-1(Question in review round 2). I don’t see a change here in any significant way. Please take out all the text that simply repeats values which can be represented with a figure, graph, or table. The text should be for discussion of the achieved results

A6. I deleted figure 8 and figure 9 (new graph) is added to discuss relationship betweeen power consumption, operating temperature and response. Contens added and revised were highlighted in blue on page 8~10. Some grammatical and typoerrors in the paper also were corrected.  

Q7. Again, same issue for the “discussion” on Figure 9, the authors just list the results from the figure in the text

Q7-1(Question in review round 2). The authors didn’t really add a significant discussion and the text that repeats all the values from Figure 9 is still in the text. Please take this out and provide a meaningful discussion. The only discussion I can see in the entire paper is simply stating that, due to the lower resistance of the type 3 heater, lower voltage can be used.

A7.  I added the discussion on figure 8 (It was figure 9 in 1st revised paper) and it is highlighted in blue on page 8~10. Some grammatical and typoerrors in the paper also were corrected.

Q8. For Figure 10, also no discussion is provided and no suggestions are made as to why this sensor seems to exhibit a higher sensitivity towards H2S. It seems to be a coincidence or a simple property of the sensing material at this particular temperature.

Q8-1(Question in review round 2) I have the same problem here as with the previous answers. The text is still predominantly repeating the numbers obtained, which are anyways presented in the graphs. And the discussion is essentially repeated for every figure in stating that the lower resistance allows to use a lower bias voltage.

A8. I added the discussion on page 10 by utilizing figure 9 and 10 and contents revised were highlighted in blue. Some grammatical and typoerrors in the paper also were corrected.

Q6. Fig. 2 shows the IDE layer above the sensor, while Fig. 3 suggests it is the other way around. The authors should clarify if they place the IDE below or above the sensing film. This will also have a large impact on the sensing reaction, since the Schottky contact between the gold and SnO2 will be modified by adsorbed gas molecules

Q6-1(Question in review round 2).But in the top view, it appears as though the sensor layer is below the heater and IDE.

A6. I revised my paper by reflecting your opinion on figure (SnO2 is depicted transparently). 
